# Physics-Based TOF Imaging Simulation for Space Targets Based on Improved Path Tracing

Zhiqiang Yan, Hongyuan Wang *, Xiang Liu, Qianhao Ning and Yinxi Lu

Space Optical Engineering Research Center, Harbin Institute of Technology, Harbin 150001, China;
18b921006@stu.hit.edu.cn (Z.Y.); 19b921006@stu.hit.edu.cn (X.L.); 19b921012@stu.hit.edu.cn (Q.N.);
20s121066@stu.hit.edu.cn (Y.L.)
* Correspondence: fountainhy@hit.edu.cn

**Abstract:** Aiming at the application of close-up space measurement based on time-of-flight (TOF) cameras, according to the analysis of the characteristics of the space background environment and the imaging characteristics of the TOF camera, a physics-based amplitude modulated continuous wave (AMCW) TOF camera imaging simulation method for space targets based on the improved path tracing is proposed. Firstly, the microfacet bidirectional reflection distribution function (BRDF) model of several typical space target surface materials is fitted according to the measured BRDF data in the TOF camera response band to make it physics-based. Secondly, an improved path tracing algorithm is developed to adapt to the TOF camera by introducing a cosine component to characterize the modulated light in the TOF camera. Then, the imaging link simulation model considering the coupling effects of the BRDF of materials, the suppression of background illumination (SBI), optical system, detector, electronic equipment, platform vibration, and noise is established, and the simulation images of the TOF camera are obtained. Finally, ground tests are carried out, and the test shows that the relative error of the grey mean, grey variance, depth mean, and depth variance is 2.59%, 3.80%, 18.29%, and 14.58%, respectively; the MSE, SSIM, and PSNR results of our method are also better than those of the reference method. The ground test results verify the correctness of the proposed simulation model, which can provide image data support for the ground test of TOF camera algorithms for space targets.

**Keywords:** TOF imaging simulation; space target; improved path tracing; BRDF; SBI

## 1. Introduction

In recent years, time-of-flight (TOF) imaging technology has been widely used in ground robot positioning and navigation, pose estimation, 3D reconstruction, indoor games, and other fields due to its advantages in structure and performance. Significantly, researchers are promoting TOF imaging technology in spatial tasks such as spatial pose estimation and relative navigation [1–6]. However, due to the particularity of the space environment, it is difficult to obtain the imaging results of the actual TOF camera before formulating the space mission scheme, planning mission content, and designing the related algorithms, so it is impossible to evaluate the algorithm capability and ensure the smooth implementation of the mission [7]. The imaging simulation method can provide data input for the back-end algorithm test of the space-based TOF camera. Nevertheless, it is different from the imaging simulation methods of the visible light camera [8–12], infrared camera [13–15], and radar [16,17], there are few imaging simulation methods of TOF cameras because the imaging principle is essentially different. Therefore, it is of great significance to develop the imaging simulation method of the TOF camera for space targets.

For the imaging simulation method of the TOF camera, the research status is as follows. References [18,19] proposed a real-time TOF simulation framework for simple geometry based on standard graphics rasterization techniques. This method only considers the

influence of errors such as dynamic motion blur and flying pixels and does not consider the influence of the background environment, so it is only suitable for indoor scenes. Reference [20] presented an amplitude modulated continuous wave (AMCW) TOF simulation method using global illumination based on the bidirectional path tracing method for indoor scenes such as kitchens. References [21,22] established physics-based TOF camera simulation methods, respectively. In order to evaluate two alternative approaches in continuous-wave TOF sensor design, reference [21] focused on realistic and practical sensor parameterization. Based on the reflective shadow map (RSM) algorithm, reference [22] introduced the bidirectional reflection distribution function (BRDF) data of materials, which has the physical imaging characteristics of actual materials, but this method also does not consider the influence of background illumination. Reference [23] proposed a pulsed TOF simulation method based on Vulkan shader and NVIDIA VKRAY ray tracing for indoor scenes. It can be seen that most of these existing TOF imaging simulation methods are only for indoor scenes, without considering the influence of background illumination, and there is no research on the imaging simulation method of TOF cameras specifically for space targets.

The main difference between the space environment and the indoor environment is the influence of sunlight. The sunlight will affect the signal-to-noise ratio and even cause the detector's supersaturation. Therefore, the camera's hardware must consider the suppression of background illumination (SBI). Many TOF detectors considering the SBI function have been developed [24–28]. At the same time, for the TOF camera simulation, SBI must also be considered, and the corresponding simulation model must be developed. Reference [29] constructed a theoretical model of SBI for PMD detectors, which can effectively characterize the detector's ability to suppress background illumination.

To sum up, this paper proposes a physics-based imaging simulation method of TOF cameras for space targets based on improved path tracing, aiming at the space application of TOF cameras. The main contributions of this method are as follows:

(1) An improved path tracing algorithm is developed to adapt to the TOF camera by introducing a cosine component to characterize the modulated light in the TOF camera.
(2) The background light suppression model is introduced, and the physics-based simulation is realized by considering the BRDF model fitted by the measured data in the near-infrared band of space materials
(3) A ground test scene is built, and the correctness of the proposed TOF camera imaging simulation method is verified by quantitative evaluation between the simulated image and measured image.

## 2. Materials and Methods

### 2.1. Imaging Principle of TOF Camera

Based on the homodyne detection principle, the AMCW TOF camera measures the distance by measuring the cross-correlation between the reflected light and reference signals. The camera first transmits the near-infrared optical signal modulated by a sine wave. The optical signal is reflected by the target surface and received by the infrared detector. The phase delay of the received signal relative to the transmitted signal is calculated to calculate the target distance information. The specific principle is shown in Figure 1.

It is assumed that the transmitted infrared optical signal is $g(t) = a\cos(2\pi f_0 t)$, its amplitude is $a$, $f_0$ is the signal modulation frequency, and the received optical signal $s(t)$ is

$$s(t) = a_r \cos(2\pi f_0 t + \varphi) + b \tag{1}$$

where $a_r$ is the amplitude of reflected signal light, $\varphi$ is the phase delay caused by target distance, and $b$ is the offset caused by ambient light. Then, the cross-correlation between the transmitted optical signal and the received optical signal is:

$$c_\tau(\varphi) = s \bigstar g = \lim_{T \to \infty} \frac{1}{T} \int_{-T/2}^{T/2} s(t)g(t+\tau)dt \tag{2}$$

where $\tau$ is the time delay and $\bigstar$ is the correlation operation symbol.

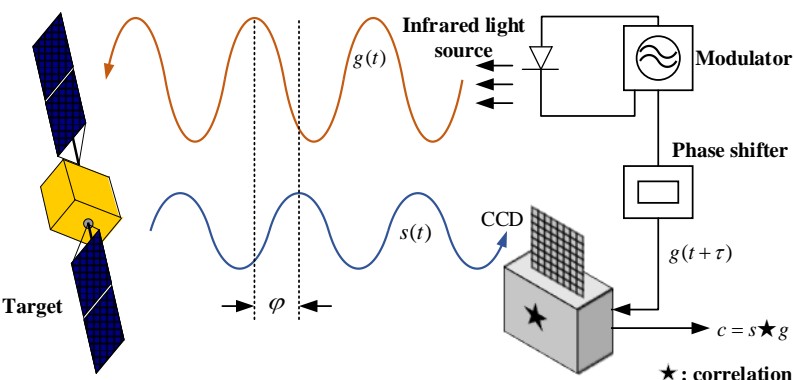

**Figure 1.** Schematic diagram of the continuous wave TOF system.

In order to recover the amplitude $a_r$ and phase $\varphi$ of the reflected light signal, four sequence amplitude images are collected generally, which are defined as:

$$C_i = c_{\tau_i}(\varphi), \quad \tau_i = i \cdot \frac{\pi}{2 \times 2\pi f_0}, \quad i \in \{0, 1, 2, 3\} \tag{3}$$

Then, the phase delay $\varphi$, reflected signal amplitude $a_r$, and offset $b$ can be obtained as:

$$\varphi = \arctan\left(\frac{C_3 - C_1}{C_0 - C_2}\right) \tag{4}$$

$$a_r = \frac{1}{2a}\sqrt{(C_3 - C_1)^2 + (C_0 - C_2)^2} \tag{5}$$

$$b = \frac{1}{4}(C_0 + C_1 + C_2 + C_3) \tag{6}$$

Finally, the distance $d$ between the TOF camera and the scene is:

$$d = \frac{1}{2}c_{light}\frac{\varphi}{2\pi f_0} \tag{7}$$

The purpose of imaging simulation is to obtain the distance $d$ and the intensity of the reflected signal, which is related to the characteristics of the target material, background, the TOF camera, and so on.

### 2.2. Imaging Characteristic Modeling

2.2.1. Target Material Characteristics Modeling

In order to achieve physics-based simulation, it is necessary to introduce the reflection characteristics of the actual material of the target surface. The BRDF is usually used to express the reflection characteristics of the material surface. As shown in Figure 2a, on the surface element $dA$, the incident light direction is $(\theta_i, \phi_i)$, and the observation direction is $(\theta_r, \phi_r)$, where $\theta$ and $\phi$ represent the zenith angle and azimuth angle, respectively, and $\hat{Z}$ represents the normal direction of the surface. The BRDF is defined as the ratio of the radiance $dL_r(\theta_i, \phi_i, \theta_r, \phi_r)$ emitted along the direction $(\theta_r, \phi_r)$ to the irradiance $dE_i(\theta_i, \phi_i)$ of the measured surface incident along the direction $(\theta_i, \phi_i)$, and the formula is as follows.

$$f_r(\theta_i, \phi_i, \theta_r, \phi_r) = \frac{dL_r(\theta_i, \phi_i, \theta_r, \phi_r)}{dE_i(\theta_i, \phi_i)} \quad (sr^{-1}) \tag{8}$$

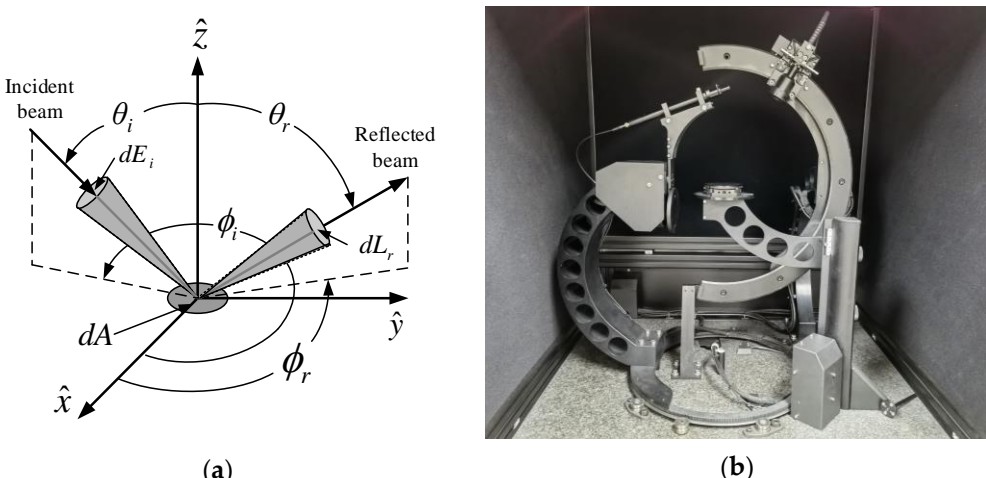

(**a**)                    (**b**)

**Figure 2.** Schematic diagram of the BRDF geometry and the measuring instrument used in this paper. (**a**) Schematic diagram of the BRDF geometry; (**b**) the REFLET-180 BRDF measuring instrument.

Yellow thermal control material and silicon solar cells are two primary surface materials of space targets. In this paper, their BRDF data are measured by the REFLET-180 BRDF measuring instrument as shown in Figure 2b. The specific material samples and some corresponding measurement results are shown in Figure 3.

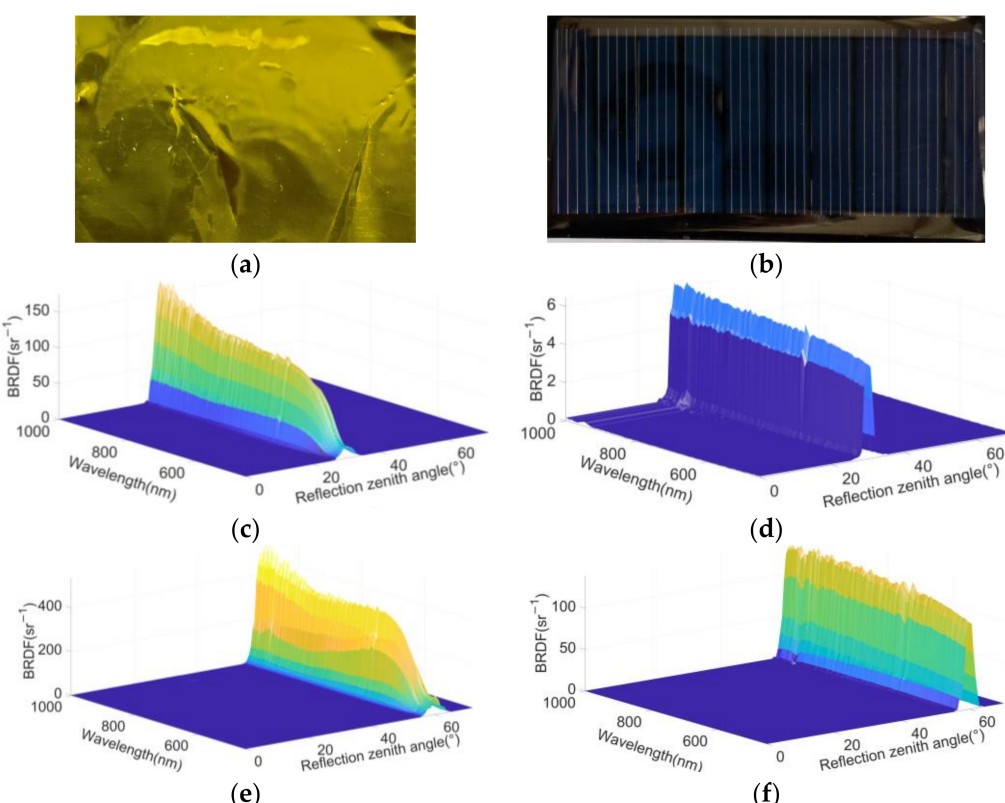

**Figure 3.** Material samples and some measured BRDF data of yellow thermal control material and silicon solar cells. The brighter the color, the greater the BRDF value. (**a**) Sample of thermal control material; (**b**) sample of silicon solar cells; (**c**) measured BRDF of thermal control material when $\theta_i$ is 30°; (**d**) measured BRDF of silicon solar cells when $\theta_i$ is 30°; (**e**) measured BRDF of thermal control material when $\theta_i$ is 60°; (**f**) measured BRDF of silicon solar cells when $\theta_i$ is 60°.

At the same time, the above measured BRDF data are theoretically modeled using the microfacet BRDF model [14,15]. The microfacet BRDF model includes the specular

reflection term and the diffuse reflection term, and the specular reflection term is the Torrance–Sparrow BRDF model [30]. The definition of the microfacet BRDF model is as follows.

$$f_r(\theta_i, \phi_i, \theta_r, \phi_r) = \frac{k_s}{4\cos\theta_i \cos\theta_r} DFG + \frac{k_d}{\pi} \tag{9}$$

where $k_s$ is the specular reflection coefficient, $k_d$ is the diffuse reflection coefficient, $D$ is the micro surface distribution factor, $F$ is the Fresnel factor [31], and $G$ is the geometric attenuation factor. Their specific definition is as follows.

$$\begin{cases} D = e^{-\tan^2\theta_h/\alpha^2} / \left(\pi\alpha^2 \cos^4\theta_h\right) \\ F = F_0 + (1 - F_0)(1 - \cos\gamma)^n \\ G = \min(1, 2\cos\theta_h \cos\theta_r/\cos\gamma, 2\cos\theta_h \cos\theta_i/\cos\gamma) \end{cases} \tag{10}$$

where $\alpha = \sqrt{2}\sigma$ and $\sigma$ are the root mean square slope of the microfacet, $F_0$ is the Fresnel coefficient at vertical incidence, and $n$ is the undetermined coefficient.

Since the wavelength of the TOF camera light source used in this paper is 850 nm, the parameters of the BRDF model at 850 nm are fitted, and the fitting error is expressed by the following formula [32,33].

$$e_{fit} = \frac{\sum\limits_{\theta_i}\sum\limits_{\theta_r}[f_{r\text{-}model} \cdot \cos(\theta_r) - f_{r\text{-}measured} \cdot \cos(\theta_r)]^2}{\sum\limits_{\theta_i}\sum\limits_{\theta_r}[f_{r\text{-}measured} \cdot \cos(\theta_r)]^2} \tag{11}$$

where $f_{r\text{-}measured}$ represents the measured BRDF value, $f_{r\text{-}model}$ represents the fitted BRDF value, the parameter fitting results are shown in Table 1, and the visualization of fitting results is shown in Figure 4.

**Table 1.** Fitting results of BRDF model parameters at 850 nm.

| | $k_s$ | $k_d$ | $n$ | $\sigma$ | $F_0$ | $e_{fit}$ |
|---|---|---|---|---|---|---|
| Thermal control material | 0.649 | 0.081 | 8.51 | 0.014 | 0.775 | 2.6% |
| Silicon solar cells | 0.350 | 0.053 | 2.22 | 0.010 | 0.021 | 5.2% |

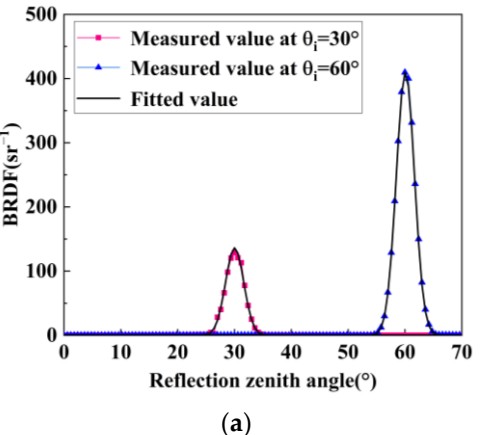
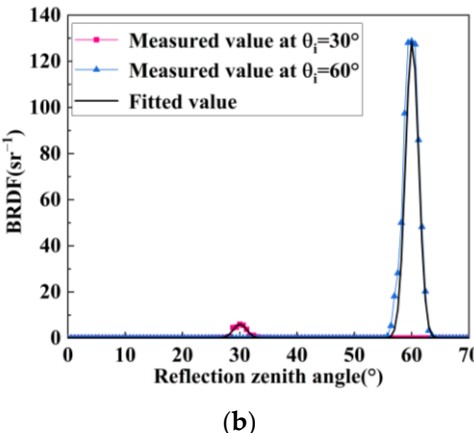

(a)                                    (b)

**Figure 4.** Measurement and model fitting results of BRDF at 850 nm for yellow thermal control material and silicon solar cells. (**a**) Fitting results of the BRDF model of yellow thermal control material; (**b**) fitting results of the BRDF model of silicon solar cells.

2.2.2. Background Characteristics Modeling

Space targets run in the Earth's orbit. The radiation in the imaging band of the TOF camera is mainly composed of direct solar radiation, solar radiation reflected by the Earth, solar radiation reflected by the Moon, and stellar radiation. Since the stellar radiation is

minimal compared to other radiation, the stellar radiation at the target can be ignored. The surface radiation of the target is shown in Figure 5.

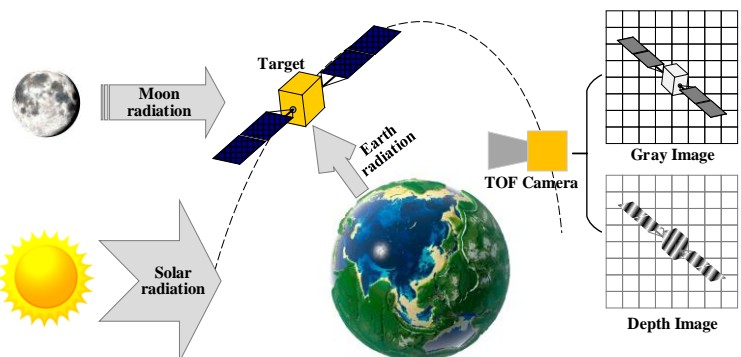

**Figure 5.** Physical radiation model of space targets.

(1)  The irradiance generated by direct solar radiation at the target is:

$$E_S = \int_{\lambda_1}^{\lambda_2} \frac{\frac{c_1}{\lambda^5}\left(e^{c_2/\lambda T_S}-1\right)^{-1} \cdot R_S^2}{R_{S-E}^2} d\lambda \tag{12}$$

where $\lambda$ is the wavelength in µm; $c_1$ is the first blackbody radiation constant; $c_2$ is the second blackbody radiation constant; $T_S$ is the solar radiation temperature, and $T_S = 5900$ K; $R_S$ is the solar radius, and $R_S = 6.5955 \times 10^5$ km; $R_{S-E}$ is the distance between the Sun and the Earth; $\lambda_1 \sim \lambda_2$ is the observation band of the TOF camera.

(2)  Assuming that the space target is in a high Earth orbit and the Earth is assumed to be a diffuse sphere, the irradiance generated by solar radiation reflected by the Earth at the target is approximate as follows.

$$\begin{aligned} E_E &= \frac{E_S \cdot \rho_E \cdot R_E^2}{\pi(R_E+H_{TE})^2} \int_{\theta_E-\frac{\pi}{2}}^{\frac{\pi}{2}} \int_{-\frac{\pi}{2}}^{\frac{\pi}{2}} \cos^3 B\left(\cos\theta_E \cos^2 l + \sin\theta_E \sin l \cos l\right) dBdl \\ &= \frac{4E_S \cdot \rho_E \cdot R_E^2}{3\pi(R_E+H_{TE})^2} \left[\cos\theta_E\left(\frac{\pi}{2} - \frac{\theta_E}{2} + \frac{\sin 2\theta_E}{4}\right) + \sin\theta_E\left(\frac{1}{4} - \frac{\cos 2\theta_E}{4}\right)\right] \end{aligned} \tag{13}$$

where $\rho_E$ is the average albedo of the Earth and $\rho_E = 0.35$; $R_E$ is the radius of the Earth and $R_E = 6370$ km; $H_{TE}$ is the height of the target from the ground; $\theta_E$ is the angle between vector $\vec{v}_{Sun\text{-}Earth}$ and vector $\vec{v}_{Earth\text{-}Target}$, and the value range is $[0, \pi]$.

(3)  Similarly, assuming that the Moon is a diffuse sphere, the irradiance generated by solar radiation reflected by the Moon at the target is:

$$\begin{aligned} E_M &\approx \frac{E_S \cdot \rho_M \cdot R_M^2}{\pi R_{TM}^2} \int_{\theta_M-\frac{\pi}{2}}^{\frac{\pi}{2}} \int_{-\frac{\pi}{2}}^{\frac{\pi}{2}} \cos^3 B\left(\cos\theta_M \cos^2 l + \sin\theta_M \sin l \cos l\right) \cdot dBdl \\ &= \frac{4E_S \cdot \rho_M \cdot R_M^2}{3\pi R_{TM}^2} \left[\cos\theta_M\left(\frac{\pi}{2} - \frac{\theta_M}{2} + \frac{\sin 2\theta_M}{4}\right) + \sin\theta_M\left(\frac{1}{4} - \frac{\cos 2\theta_M}{4}\right)\right] \end{aligned} \tag{14}$$

where $\rho_M$ is the average albedo of the Moon and $\rho_M = 0.12$; $R_M$ is the radius of the Moon and $R_M = 1738$ km; $R_{TM}$ is the distance between the target and the center of mass of the Moon; $\theta_M$ is the angle between vector $\vec{v}_{Sun\text{-}Moon}$ and vector $\vec{v}_{Moon\text{-}Target}$, and the value range is $[0, \pi]$.

### 2.2.3. SBI Characteristics Modeling

The TOF sensor is commonly the photonic mixer device (PMD). The PMD is a two-tap sensor, and the structure diagram is shown in Figure 6a [34]. The interference of background light on the actively modulated light leads to the premature saturation of the quantum

well of the pixel, so that less reflected active light containing depth information is detected, which will lead to increased noise and a reduced signal-to-noise ratio (SNR). The schematic diagram of the influence of background light on PMD is shown in Figure 6b [35]. Therefore, designing a system that can effectively reduce the impact of background illumination is an important job.

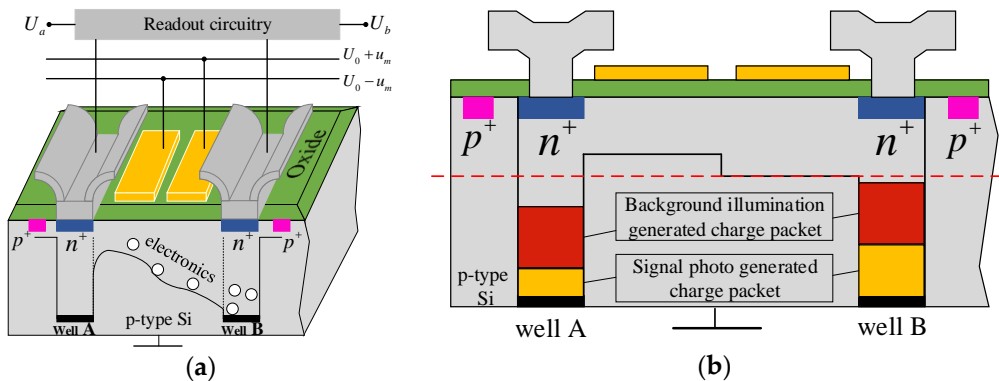

**Figure 6.** The structure diagram of the PMD and the schematic diagram of the influence of background light on the PMD. (**a**) The structure diagram of the PMD; (**b**) the schematic diagram of the influence of background illumination on the PMD.

The suppression of background illumination (SBI) developed by PMDTec has been successfully applied to TOF cameras such as CamCube. The SBI is an in-pixel circuitry that subtracts ambient light, which prevents the pixels from saturating. The manufacturer did not publish the specific details of the SBI compensation circuit, but reference [36] summarizes the response relationship of the A and B channels of the PMD pixel to the exposure time through experiments, as shown in Figure 7. The PMD pixel will produce a photoelectric response to both the signal light and the background illumination during the exposure time. In the linear region, the SBI is not activated. When the number of electrons in the quantum well A or B reaches $n_{SBI,start}$, the SBI is activated. As the exposure time continues to increase, it will enter the SBI Limit region, and the data of this pixel is invalid at this time.

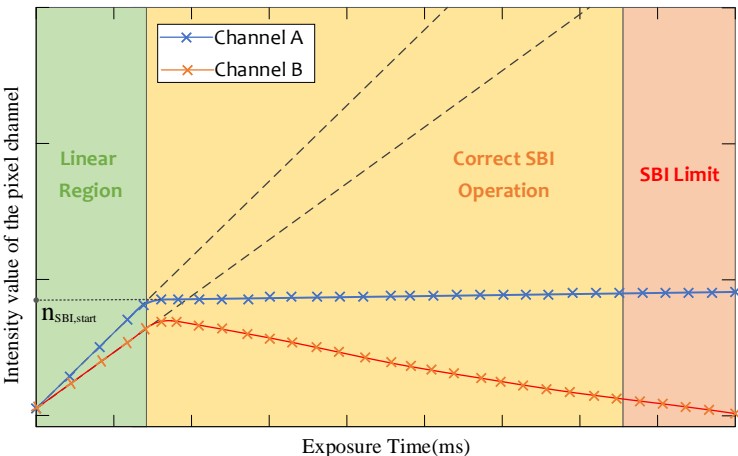

**Figure 7.** The response relationship of the A and B channels of the PMD pixel to the exposure time.

Reference [29] established the SBI model shown in Figure 8 by analyzing the response relationship of the TOF camera to illumination intensity or exposure time. This SBI model is also used in this paper, which can be described as follows. The charge stored in two quantum wells $\sum$ is continuously compared with a reference value $n_{SBI,start}$, and once the stored charge in one of the quantum wells exceeds this value, that is, the difference $n_{\Delta}$

between the stored charge and the reference value $n_{SBI,start}$ is positive, a compensation process will be triggered. Two compensation currents are injected into the two quantum wells during the compensation process, respectively, and the number of charges contained in the compensation currents is approximately the same as that of $n_\Delta$. After compensation, the quantum well containing more electrons is reset to $n_{SBI,start}$, and the charge of the other quantum well is set to a value lower than $n_{SBI,start}$.

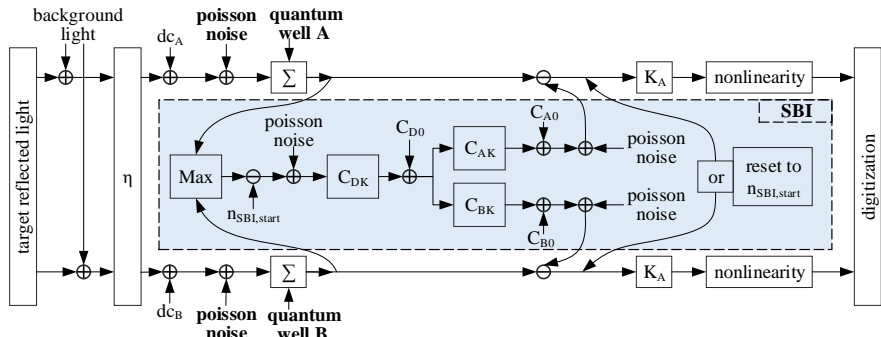

**Figure 8.** The SBI model. $K_A$ and $K_B$ represent the charge-to-voltage coefficient. $C_{DK}$ and $C_{D0}$ represent the charge difference coefficient and offset, respectively. (or $C_{BK}$) and $C_{A0}$ (or $C_{B0}$) represent the compensation charge coefficient and offset, respectively.

The compensation process does not lose any critical information, as the phase delay can be estimated by keeping only the difference between the charges in the two quantum wells. As shown below, each sub-amplitude image in Equation (4) is the difference between the amplitude images of channels A and B.

$$C_i = C_{iA} - C_{iB} \tag{15}$$

where $C_{iA}$ and $C_{iB}$ are amplitude images output by channels A and B, respectively.

As shown in the figure above, both the background illumination and the SBI process introduce additional Poisson noise. The Poisson distribution is given by

$$P_\lambda = \frac{\lambda^k}{k!} e^{-\lambda} \tag{16}$$

where $\lambda$ describes the mean of the values, which is here the number of generated electrons. $P_\lambda$ is the probability of detecting $k$ electrons for a given $\lambda$.

### 2.3. Imaging Simulation Modeling

In order to obtain the simulated image of the actual target, an AMCW TOF camera imaging simulation model based on path tracing is proposed in this paper. Firstly, establish the space three-dimensional target scene, simulate the TOF camera to transmit the modulated light signal to the target scene, and obtain the radiance image of the space target scene through the path tracing method. Secondly, the radiance image is coupled with the imaging chain factors such as platform motion, optical system, and detector, and the influence of background light is suppressed through the SBI module. Then, four frames of amplitude images are obtained through time sampling. Finally, the final depth and grey image are obtained through Formula (4) to Formula (7). The specific simulation model is shown in Figure 9.

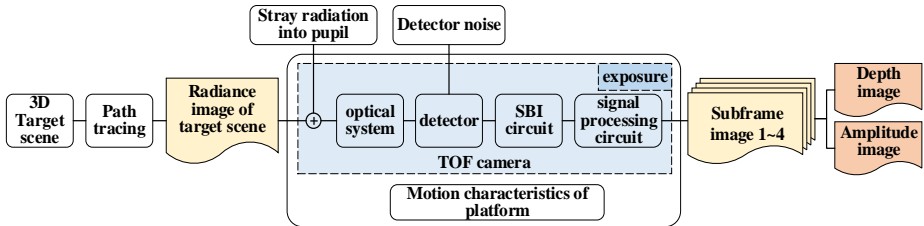

**Figure 9.** AMCW TOF camera imaging simulation model.

2.3.1. Improved Path Tracing Algorithm of the TOF Camera

The general path tracing algorithm is suitable for the imaging simulation of visible light and infrared cameras. However, since the light signal in the TOF camera is modulated, there is a cosine component that varies with the propagation distance. Therefore, to apply the path tracing algorithm to the TOF camera, an improved path tracing algorithm is developed by introducing a cosine component to characterize the modulated light in the TOF camera, and the improved algorithm is shown as follows.

Define $T(g(t), d)$ to represent the propagation distance $d$ [m] of the optical signal $g(t) = b + a\cos(2\pi f_0 t) = g(t)^- + real(g(t)^{\cos})$ without attenuation, and its form is as follows.

$$
\begin{aligned}
T(g(t), d) &= b + a\cos\left(2\pi f_0 t + 2\pi f_0 \tfrac{d}{c}\right) \\
&= g(t)^- + real\left(g(t)^{\cos} \cdot e^{i\Psi d}\right)
\end{aligned}
\tag{17}
$$

where $g(t)^- = b$ represents the DC component, $g(t)^{\cos} = ae^{i(2\pi f_0 t)}$ represents the cosine component, and $real()$ represents the real part of the imaginary number and $\Psi = \frac{2\pi f_0}{c}$.

The light source $L$ of the TOF camera is assumed to be a uniform point light source, then the light intensity $I_L$ is as follows.

$$
I_L(t) = \frac{P_L}{\Omega_L}[1 + \cos(2\pi f_0 t)]
\tag{18}
$$

where $P_L$[W] represents the power of the light, $\Omega_L$ represents the solid angle, and $f_0$ represents the modulation frequency of the optical signal.

The illuminance $E_L(t, d)$ at the micro-plane perpendicular to the light propagation direction at $d$ [m] away from the light source is:

$$
E_L(t, d) = T\left(I_L(t)/d^2, d\right)
\tag{19}
$$

For a point $P$ in the scene shown in Figure 10, the illuminance generated by direct lighting at $P$ is $E_{L\to@P}(t, d_{L\to P})$.

$$
E_{L\to@P}(t, d_{L\to P}) = E_L(t, d_{L\to P}) \cdot \cos\theta_{P\to L}
\tag{20}
$$

where $d_{L\to P}$ is the distance between the light source $L$ and the point $P$, and $\theta_{P\to L}$ is the included angle between the vector $\vec{v}_{P\to L}$ of $P$ pointing to $L$ and the normal vector $\vec{n}_P$ of the surface at point $P$.

The direct illumination radiance $L^{direct}_{@P\to S}(t)$ generated by the infrared light source along the direction of $P \to S$ at point $P$ is:

$$
L^{direct}_{@P\to S}(t) = E_{L\to@P}(t, d_{L\to P}) \cdot f_{L\to P\to S}
\tag{21}
$$

where $f_{L\to P\to S}$ represents the BRDF of the surface. For the irradiance $E_{L'\to@P}(t, d_{L'\to P})$ of ambient light sources $L'$ such as the Sun, the Earth, and the Moon, as shown in Formulas (12)–(14), the cosine component of $E_{L'\to@P}(t, d_{L'\to P})$ is 0, only the DC component. Therefore, in the derivation process, this paper only gives the case of the active light of

the TOF camera. When encountering ambient light, it only needs to change the cosine component related to ambient light to 0.

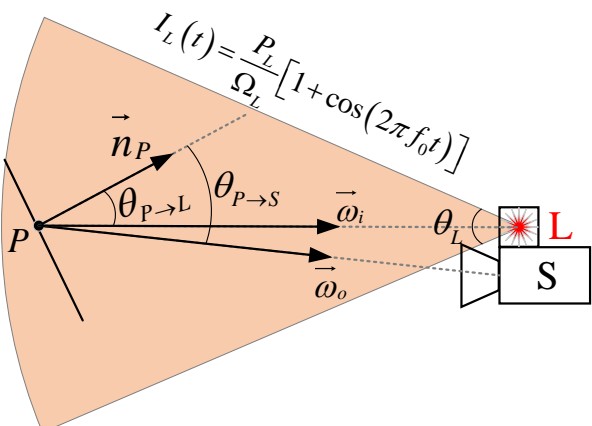

**Figure 10.** The schematic diagram of direct lighting.

In addition to the direct illumination of light sources, the light signal reflected by other surfaces will also affect the radiance at point $P$, as shown in Figure 11. We call this part indirect illumination radiance denoted by $L_{@P \to S}^{indirect}(t)$, which is defined as follows.

$$L_{@P \to S}^{indirect}(t) = \int_{\Omega} f_{P' \to P \to S} L_{P' \to @P}(t) \cos(\theta_i) d\varpi_i \tag{22}$$

where $L_{P' \to @P}(t)$ represents the radiance generated by $P'$ at point $P$, $\Omega$ represents the solid angle of surfaces that contribute to the radiance at point $P$, $\varpi_i$ represents the micro solid angle in the $\vec{\omega}_i$ direction, and $d\varpi_i$ is defined as follows.

$$d\varpi_i = \frac{\cos(\theta_o)dA(P')}{d_{P' \to P}^2} \tag{23}$$

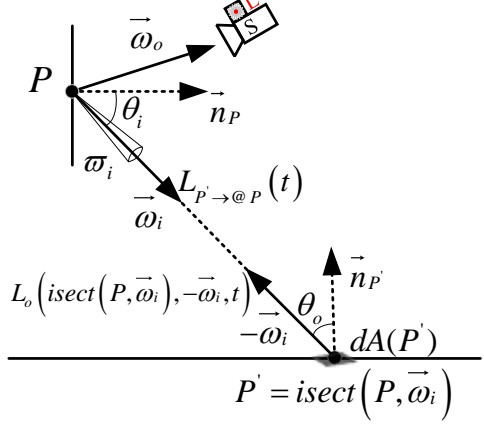

**Figure 11.** Radiative transfer between two points.

Since the radiation energy of light in the scene is conserved, the radiance at $P$ can be associated with the radiance at another point $P'$.

$$L_{P' \to @P}(t) = T\left(L_o\left(isect\left(P, \vec{\omega}_i\right), -\vec{\omega}_i, t\right), d_{P' \to P}\right) \tag{24}$$

where $isect\left(P, \vec{\omega}_i\right)$ calculates the first intersection $P'$ between the light propagating along the $\vec{\omega}_i$ direction from point $P$ and the scene. $dA(P')$ represents the microfacet at the intersection $P'$.

Substituting Equation (23) into Equation (22), we can get:

$$L_{@P \to S}^{indirect}(t) = \int_A f_{P' \to P \to S} L_{P' \to @P}(t) \frac{V(P \leftrightarrow P')|\cos(\theta_i)||\cos(\theta_o)|}{d_{P' \to P}^2} dA(P') \tag{25}$$

where $A$ is all surfaces of the scene, and $V(P \leftrightarrow P')$ represents the visibility between points $P'$ and $P$. If the two points are visible to each other, $V(P \leftrightarrow P') = 1$; otherwise, $V(P \leftrightarrow P') = 0$ Let.

$$G(P \leftrightarrow P') = \frac{V(P \leftrightarrow P')|\cos(\theta_i)||\cos(\theta_o)|}{d_{P' \to P}^2} \tag{26}$$

Then

$$L_{@P \to S}^{indirect}(t) = \int_A f_{P' \to P \to S} L_{P' \to @P}(t) G(P \leftrightarrow P') dA(P') \tag{27}$$

Combining direct radiance and indirect radiance, the radiance $L_{@P \to S}(t)$ at the point $P$ can be expressed as follows.

$$\begin{aligned} L_{@P \to S}(t) &= L_{@P \to S}^{direct}(t) + L_{@P \to S}^{indirect}(t) \\ &= E_{L \to @P}(t, d_{L \to P}) \cdot f_{L \to P \to S} + \int_A f_{P' \to P \to S} L_{P' \to @P}(t) G(P \leftrightarrow P') dA(P') \end{aligned} \tag{28}$$

After the radiance $L_{@P \to S}(t)$ propagate distance $d_{P \to S}$, the radiance $L_{P \to @S}(t)$ at the camera sensor $S$ is:

$$L_{P \to @S}(t) = T(L_{@P \to S}(t), d_{P \to S}) \tag{29}$$

The path tracing algorithm [37,38] based on Monte Carlo is utilized to solve the Equation (28), which is approximated as follows.

$$L_{@P_1 \to S}(t) \approx \frac{1}{N} \sum_{k=1}^{N} \sum_{l}^{NumLight} \sum_{n=1}^{MaxDepth} L_{MC}^k(\overline{p}_n) \tag{30}$$

where $MaxDepth$ represents the maximum bounce times of light, $NumLight$ represents the number of light sources, $N$ represents the number of samples per pixel, $L_{MC}^k(\overline{p}_n)$ represents the Monte Carlo estimation of the $k$th sample, as shown below.

$$\begin{aligned} L_{MC}(\overline{p}_n) = real \Bigg\{ & \frac{\bar{E}_{L \to @P_n}(t, d_{L \to P_n}) \cdot f_{L \to P_n \to S} \cdot f_{P_n \to P_{n-1} \to P_{n-2}} G(P_n \leftrightarrow P_{n-1})}{p_A(P_n)} \\ & \times \left( \prod_{i=1}^{n-2} \frac{f_{P_{i+1} \to P_i \to P_{i-1}} \cdot V(P_i \to P_{i+1}) \cdot |\cos \theta_i|}{p_\omega(P_i \to P_{i+1})} \right) \\ & + \frac{E_{L \to @P_n}^{\cos}(t, d_{L \to P_n}) \cdot f_{L \to P_n \to S} \cdot e^{i\Psi d_{P_n \to P_{n-1}}} f_{P_n \to P_{n-1} \to P_{n-2}} G(P_n \leftrightarrow P_{n-1})}{p_A(P_n)} \\ & \times \left( \prod_{i=1}^{n-2} \frac{e^{i\Psi d_{P_{i+1} \to P_i}} \cdot f_{P_{i+1} \to P_i \to P_{i-1}} \cdot V(P_i \to P_{i+1}) \cdot |\cos \theta_i|}{p_\omega(P_i \to P_{i+1})} \right) \Bigg\} \end{aligned} \tag{31}$$

where $p_A(P_n)$ is the sampling probability function at $P_n$ point, which is defined as $p_A(P_n) = \frac{1}{\sum_n A_n}$, $\sum_n A_n$ is the sum of the surfaces of the scene, and $p_\omega(P_i \to P_{i+1})$ is the sampling probability density function in the $P_i \to P_{i+1}$ direction.

The irradiance $E_{pixel}$ received by the pixel of the TOF sensor is as follows.

$$E_{pixel}(t) = T(L_{@P_1 \to S}(t), d_{P_1 \to S}) \cdot \tau \cdot \frac{\pi}{4} \cdot \left( \frac{1}{\text{F/\#}} \right)^2 \cos^4 \alpha \tag{32}$$

where, $\tau$ represents the transmittance of the AMCW TOF camera, F/# represents its f-number, and $\alpha$ represents the angle between the optical axis and the vector $\overrightarrow{v}_{S \to P_1}$.

2.3.2. Imaging Link Impact Modeling

As shown in Figure 9, the target scene radiation forms the conversion process of "radiation-voltage-grey" after passing through each component module of the TOF camera. In this process, the signal is affected by the optical system, detector, circuit processing unit, and platform, which is reflected in the image as the dispersion effect on the radiation image. These dispersion effects can be described by each module's modulation transmission function (MTF), as shown in Figure 12.

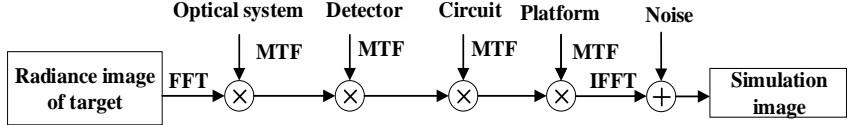

**Figure 12.** Mathematical model of the radiative transfer process.

Then, the last output simulation image [39] is:

$$Image_{sim} = IFFT(FFT(Image_{Rad}) \cdot MTF_{opt} \cdot MTF_{\det} \cdot MTF_e \cdot MTF_{mot}) + Noise \tag{33}$$

where $FFT$ represents fast Fourier transform; $MTF_{opt}$ represents the MTF of the optical system; $MTF_{\det}$ represents the MTF of the detector; $MTF_e$ represents the MTF of the signal processing circuit; $MTF_{mot}$ represents the MTF of the platform motion; *Noise* represents the noise of the image. MTF models of different processes can be found in references [8,17,40].

## 3. Results

In order to verify the correctness of the proposed TOF camera imaging simulation method for space targets, the ground experiment scene shown in Figure 13 is built. Considering that it is difficult to simulate the radiation of the Earth and the Moon on the ground, and the influence of these radiations is minimal, the ground experiment only considers the direct solar radiation. The experimental scene comprises a satellite model, a TOF camera, a turntable, a black background, and a solar simulator. The surface of the satellite model is mainly composed of yellow thermal control materials and silicon solar cells, and the TOF camera is placed horizontally on a tripod. The TOF camera used in this paper is the PMD CamBoard camera, and its performance indexes are shown in Table 2. The solar simulator used is the Newport Oriel Sol3A, and its performance indexes are shown in Table 3. The experimental background is a dark background composed of black light-absorbing flannel, and the light beam emitted by the solar simulator completely covers the target model.

During the experiment, all light sources in the room were turned off, the solar simulator was turned on, the output power of the solar simulator was set to the typical power (1 SUN), and the PMD camera was used to capture the images of the target model. Due to the size of the satellite model and the limited illumination area of the solar simulator, to make the beam of the solar simulator cover the target model entirely, the satellite model is inclined at a certain angle. At the same time, the TOF imaging simulation method proposed in this paper was used to obtain the corresponding simulation image according to the experimental conditions. Other simulation parameters used are shown in Table 4. The measured images and the simulated images are shown in Figure 14. In order to facilitate the comparison with the simulated image, the background area of the measured image is eliminated manually.

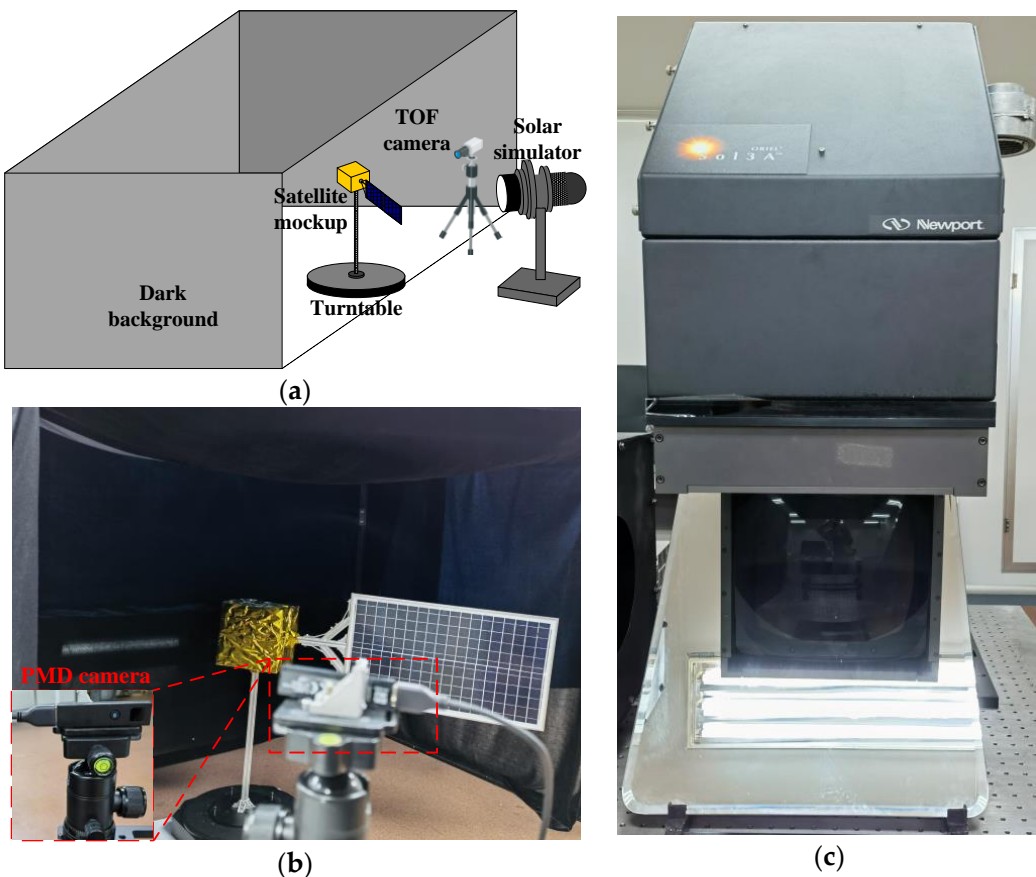

**Figure 13.** Ground experiment scene. (**a**) Schematic diagram of the ground experiment scene; (**b**) actual ground experiment scene; (**c**) the Newport Oriel Sol3A Solar Simulator, Model 94123A.

**Table 2.** The performance indexes of PMD CamBoard.

| Indexes | Value |
|---|---|
| Resolution | 224 × 171 pixel |
| Wavelength of light source | 850 nm |
| Field angle | 62° × 45° |
| Focal length ($f/dx$, $f/dy$) | (208.33, 208.33) |
| Aperture | 2 mm |
| Acquisition time per frame | 4.8 ms typical at 45 fps |
| Average power consumption | 300 mW |

**Table 3.** Some performance indexes of the Newport Oriel Sol3A Solar Simulator, Model 94123A.

| Indexes | Value |
|---|---|
| Illumination area | 305 mm × 305 mm |
| Maximum angle of incidence | (half angle) < ±0.5° |
| Typical power output | 100 mW/cm$^2$ (1 SUN), ±20% Adjustable |
| Uniformity | <±2% |
| Spectral match | 9.7–16.1% (800–900 nm) |

**Table 4.** Other simulation parameters used.

| Indexes | Value |
|---|---|
| Size of satellite body | $20 \times 20 \times 20$ cm |
| Size of solar panel | $63 \times 35$ cm |
| $d_{cam\text{-}sat}$ | 1.5 m |
| $\theta_{cam\text{-}panel}$ | $60°$ |
| $\theta_{solarSim\text{-}panel}$ | $10°$ |
| $n_{SBI,start}$ | 36,500 |

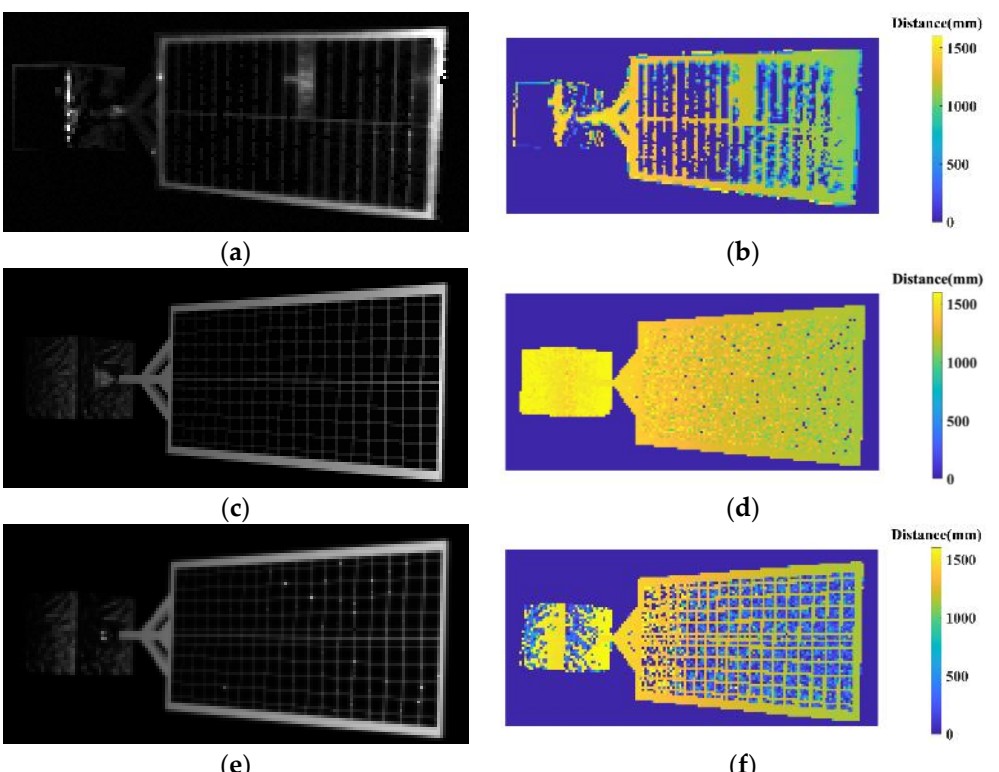

(a)  (b)

(c)  (d)

(e)  (f)

**Figure 14.** The measured images of the ground experiment scene (the background area is eliminated) and the corresponding simulated images. (**a**) Measured grey image; (**b**) measured depth image; (**c**) grey image simulated by the method of Ref. [22]; (**d**) depth image simulated by the method of Ref. [22]; (**e**) simulated grey image of ours; (**f**) simulated depth image of ours. (**c**,**d**) only consider the influence of the active light of the TOF camera and do not consider the ambient solar light. (**e**,**f**) consider the process of the SBI model suppressing solar ambient illumination.

In Table 4, $d_{cam\text{-}sat}$ represents the distance between the PMD TOF camera and satellite body center, $\theta_{cam\text{-}panel}$ represents the angle between the optical axis of the camera and the plane of the solar panel, and $\theta_{solarSim\text{-}panel}$ represents the angle between the beam of the solar simulator and the plane of the solar panel. According to references [41,42], the value of $n_{SBI,start}$ is about 36,500.

In order to quantitatively evaluate the simulation effect, the mean, variance, mean square error (MSE), structural similarity (SSIM), and peak signal-to-noise ratio (PSNR) of the grey image and depth image are calculated. The smaller the MSE, the larger the SSIM, and the larger the PSNR, indicating the more realistic the simulation results are. These indexes are shown in Table 5, where bold indicates better results. At the same time, the depth image is converted into a point cloud and denoised. The point cloud before and after denoising is shown in Figure 15.

**Table 5.** Comparison of indexes between measured images and simulated images.

| Index | | Measured Mage | Ref. [22]'s Results | Our Results | Ref. [22]'s Error | Our Error |
|---|---|---|---|---|---|---|
| Grey | Mean | 17.79 | 16.73 | 18.25 | 5.96% | **2.59%** |
| | Var | 1411.67 | 1347.53 | 1358.06 | 4.54% | **3.80%** |
| | MSE | — | 1788.35 | **1782.60** | — | — |
| | SSIM | — | 0.70 | **0.72** | — | — |
| | PSNR | — | 15.60 | **15.62** | — | — |
| Depth | Mean | 403.04 | 750.28 | 476.75 | 86.16% | **18.29%** |
| | Var | $2.95 \times 10^5$ | $4.15 \times 10^5$ | $3.38 \times 10^5$ | 40.68% | **14.58%** |
| | MSE | — | $5.46 \times 10^5$ | $\mathbf{4.45 \times 10^5}$ | — | — |
| | SSIM | — | 0.80 | **0.85** | — | — |
| | PSNR | — | 38.95 | **39.85** | — | — |

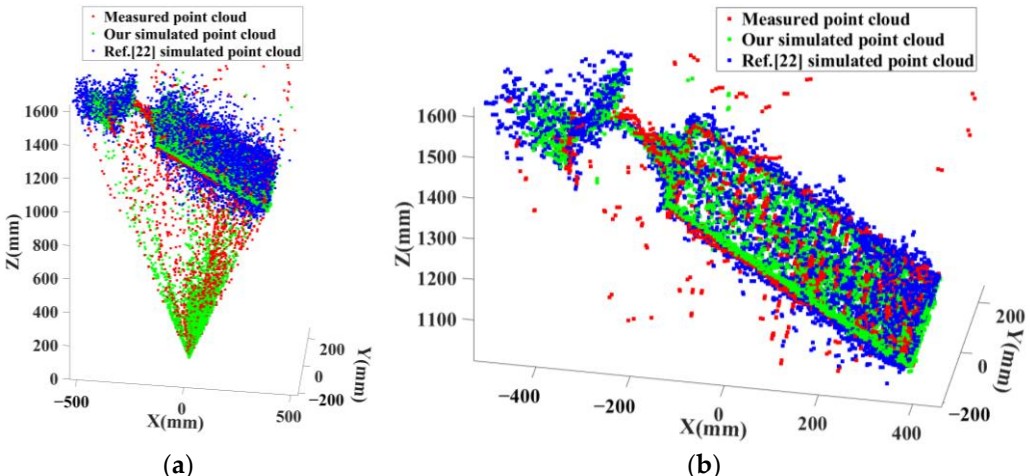

**Figure 15.** The simulated [22] and measured point cloud. (**a**) Original simulated and measured point clouds; (**b**) simulation and measured point cloud after denoising.

## 4. Discussion

As shown in Figure 14, our simulated grey and depth images are very similar to the measured grey and simulation images in visual effect. The gray image simulated by the method of reference [22] is also very similar to the measured gray image. However, the corresponding simulated depth image is quite different from the measured depth. The specific reason is that this method does not consider the error introduced by background lighting, resulting in low depth noise. At the same time, as shown in Figure 15, the point cloud coincidence degree of our method is better than that of reference [22]. In addition, it can be seen from Table 5 that the relative error of the grey mean, grey variance, depth mean, and depth variance is 2.59%, 3.80%, 18.29%, and 14.58%, respectively, which is better than the results of the method in reference [22]. Furthermore, our method's MSE, SSIM, and PSNR results are also better than those of the method in reference [22].

## 5. Conclusions

This paper proposed a physics-based AMCW TOF camera imaging simulation method based on the improved path tracing for space targets based on the analysis of space background environment characteristics and the TOF camera imaging mechanism. Firstly, the BRDF data of the yellow thermal control material and the silicon cell at 850 nm was measured, the parameters of the microfacet BRDF model were fitted, and the fitting error was less than 5.2%. Secondly, an improved path tracing algorithm was developed to adapt to the TOF camera by introducing a cosine component to characterize the modulated light in the TOF camera. Then, the imaging link simulation model considering the coupling effects

of the BRDF of materials, SBI, optical system, detector, electronic equipment, platform vibration, and noise was established. Finally, the ground experiment was carried out, and the relative error of the grey mean, grey variance, depth mean, and depth variance was 2.59%, 3.80%, 18.29%, and 14.58%, respectively. At the same time, our method's MSE, SSIM, and PSNR results were also better than those of the reference method. The experimental results verify the correctness of the proposed simulation method and can provide image data support for the ground test of TOF camera algorithms for space targets.

**Author Contributions:** Conceptualization, H.W.; methodology, Z.Y.; software, Z.Y.; validation, Z.Y., X.L., Q.N. and Y.L.; formal analysis, Z.Y.; investigation, Z.Y., X.L. and Q.N.; writing—original draft preparation, Z.Y.; writing—review and editing, Z.Y. and H.W.; visualization, Z.Y., X.L. and Q.N.; supervision, H.W.; project administration, H.W.; funding acquisition, H.W. All authors have read and agreed to the published version of the manuscript.

**Funding:** This research was funded by National Natural Science Foundation of China, grant number 61705220.

**Data Availability Statement:** Not applicable.

**Acknowledgments:** We sincerely thank the National Natural Science Foundation of China for its support.

**Conflicts of Interest:** The authors declare no conflict of interest.

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
