# Peer review of "Physics-Based TOF Imaging Simulation for Space Targets Based on Improved Path Tracing"

_remotesensing, doi:10.3390/rs14122868_

Round 1

Reviewer 1 Report

The authors propose in this paper a physics-based amplitude modulated continuous wave TOF (time-of-flight) camera imaging simulation method for space targets based on original accurate path tracing. Their argument in dealing with the subject is the stringent need for a better tool for imaging simulation of space targets, as current methods based on visible light camera, infrared camera or radar are far from being efficient. The recently proposed TOF-camera based imaging simulation method seems to meet researchers’ expectations. The few papers published on the subject until now are critically reviewed. The proposed method is an improved implementation of this novel principle.

For the beginning, the authors present shortly and very clear the imaging principle of TOF camera based on amplitude modulated continuous wave, synthesize the associated analytics and derive the expression for the distance separating the TOF camera and the target. Then, efforts are made for imaging characteristic specific modeling. They consist in target material characteristic modeling, namely the yellow thermal control material and the silicon solar cells, for which model parameters are derived by curve fitting, and suppression background interference (SBI) modeling. For the latter case, the authors took the commonly used photonic mixer device and illustrate the premature saturation of the quantum well of the pixel by the background light and use an established technique to avoid this by means of compensation currents.

The proposed imaging simulation method consists in a set of successive steps, synthetically presented in a block diagram: establishing the 3D target scene, simulating the TOF camera to transmit the modulated light signal to the target, obtaining the radiance image of the target scene through the path tracing method, coupling the radiance image with the imaging chain factors such as platform motion, optical system, and detector, and suppressing the background interference. Finally, the depth and the grey image are obtained by means of specific formulas. The main contribution claimed by the authors is the improvement of the path tracing algorithm brought by the introduction of a cosine component to characterize the modulated light in the TOF camera. This improvement is proved analytically.

The performance of the proposed method is checked by means of an ground experimental set up consisting in a satellite model covered with yellow thermal control materials and silicon solar cells. A solar simulator is used to generate a Sun-like illumination, while Earth and Moon radiation is neglected.

The measured images of the ground experiment, the simulated images by the proposed method, and the simulated images form a reference paper are comparatively presented. The authors claim that their method offers the best values for the average and the variance for the depth and the grey image.

The subject is clearly presented, the authors use adequate analytical tools to sustain their arguments, the experiment is detailly described, and the results are thoroughly commented. The conclusions are consistent with paper content and the reported results.

The references in the list are adequate.

Author Response

Dear Reviewers:

Thank you for your letter and the comments concerning our manuscript entitled "Physics-based TOF imaging simulation for space targets based on improved path tracing" (ID: remotesensing-1749121). Those comments are all valuable and very helpful for revising and improving our paper, as well as the essential guiding significance to our research. We have studied the comments carefully and have made a correction which we hope meets with approval. Revised portions are marked up using the “Track Changes” function. The paper's primary corrections and response to the reviewer's comments are as follows.

Responses to Reviewer 1

Thank you sincerely for your letter and the comments and for your approval of our manuscript.

Reviewer 2 Report

In this paper a new method for simulation of camera imaging is proposed. The target camera is a time-of-flight camera, physics-based simulation method includes fitting of the BRDF of a typical space target. This simulation methods helps to get imaging results of the actual TOF camera before having the space mission, and helps in formulating scheme, planning the mission content, test the back-end algorithms for space-based TOF camera. The paper describes in detail the imaging principles of TOF camera, the algorithm of fitting BRDF, improved path-tracing algorithm for more precise measurements. Verification of the proposed technique is done using experimental set up with model of space target, space light source (solar simulator), dark background (light-absorbing flannel). The result provide good agreement between measured and simulated images, the result are also compared to the existing solution, and show superiority of the proposed technique. The simulation method can provide the data necessary for ground tests of TOF camera algorithms for space targets.

Author Response

Dear Reviewers:

Thank you for your letter and the comments concerning our manuscript entitled "Physics-based TOF imaging simulation for space targets based on improved path tracing" (ID: remotesensing-1749121). Those comments are all valuable and very helpful for revising and improving our paper, as well as the essential guiding significance to our research. We have studied the comments carefully and have made a correction which we hope meets with approval. Revised portions are marked up using the “Track Changes” function. The paper's primary corrections and response to the reviewer's comments are as follows.

Responses to Reviewer 2

Thank you sincerely for your letter and the comments and for your approval of our manuscript.
